# Computationally designed GPCR quaternary structures bias signaling pathway activation

Justine S. Paradis[1,2,7], Xiang Feng [3,6,7], Brigitte Murat[1,2], Robert E. Jefferson [3], Badr Sokrat[1,2], Martyna Szpakowska [4], Mireille Hogue[2], Nick D. Bergkamp[5], Franziska M. Heydenreich[1,2], Martine J. Smit [5], Andy Chevigné [4], Michel Bouvier [1,2,8] ✉ & Patrick Barth [3,8] ✉

Communication across membranes controls critical cellular processes and is achieved by receptors translating extracellular signals into selective cytoplasmic responses. While receptor tertiary structures can be readily characterized, receptor associations into quaternary structures are challenging to study and their implications in signal transduction remain poorly understood. Here, we report a computational approach for predicting receptor self-associations, and designing receptor oligomers with various quaternary structures and signaling properties. Using this approach, we designed chemokine receptor CXCR4 dimers with reprogrammed binding interactions, conformations, and abilities to activate distinct intracellular signaling proteins. In agreement with our predictions, the designed CXCR4s dimerized through distinct conformations and displayed different quaternary structural changes upon activation. Consistent with the active state models, all engineered CXCR4 oligomers activated the G protein Gi, but only specific dimer structures also recruited β-arrestins. Overall, we demonstrate that quaternary structures represent an important unforeseen mechanism of receptor biased signaling and reveal the existence of a bias switch at the dimer interface of several G protein-coupled receptors including CXCR4, mu-Opioid and type-2 Vasopressin receptors that selectively control the activation of G proteins vs β-arrestin-mediated pathways. The approach should prove useful for predicting and designing receptor associations to uncover and reprogram selective cellular signaling functions.

A wide range of membrane proteins, including single-pass receptor tyrosine kinases, cytokine receptors, and ion channels, functions through the folding and association of several polypeptide chains into specific quaternary structures. The functional role of oligomerization in other membrane protein classes remains controversial as the observation of receptor associations is very sensitive to the experimental conditions and techniques[1–4]. Receptors from the largest class of G protein-coupled receptors (GPCRs) were often observed as

[1]Department of Biochemistry and Molecular Medicine, Université de Montréal, Montréal, QC H3T 1J4, Canada. [2]Institute for Research in Immunology and Cancer (IRIC), Université de Montréal, Montréal, QC H3T 1J4, Canada. [3]Interfaculty Institute of Bioengineering, Ecole Polytechnique Fédérale de Lausanne, Lausanne CH-1015, Switzerland. [4]Department of Infection and Immunity, Immuno-Pharmacology and Interactomics, Luxembourg Institute of Health, Esch-sur-Alzette, Luxembourg. [5]Amsterdam Institute for Molecules, Medicines and Systems (AIMMS), Division of Medicinal Chemistry, Faculty of Sciences, Vrije Universiteit, Amsterdam, The Netherlands. [6]Present address: Department of Structural Biology, Van Andel Institute, Grand Rapids, MI, USA. [7]These authors contributed equally: Justine S. Paradis, Xiang Feng. [8]These authors jointly supervised this work: Michel Bouvier, Patrick Barth. ✉e-mail: michel.bouvier@umontreal.ca; patrick.barth@epfl.ch

oligomers in electron microscopy, X-ray crystallography, and BRET studies[5–11]. However, when trapped as monomers in nanolipid disks, GPCRs, such as rhodopsin and β2 adrenergic receptor, remained functional, binding and activating their primary intracellular signaling G proteins[12,13]. Structural and biochemical studies suggested that different GPCRs can self-associate through distinct transmembrane helical (TMH) interfaces. Computational modeling approaches based on molecular dynamics simulations have also identified different possible modes and lifetimes of GPCR associations[14,15] but the functional relevance of these oligomeric forms remains poorly understood[5–7,16–22]. For example, chemokine receptor CXCR4 signaling is linked to the formation of nanoclusters at the cell membrane[23]. Such nanoclusters are controlled by key structural motifs present at the receptor TMH surface but do not involve the receptor dimeric interface observed in X-ray structures[10].

In principle, computational protein design techniques can probe and decipher the importance of protein associations by reprogramming protein-protein interactions or designing competitive binding inhibitors, but these approaches have mostly been applied to soluble proteins[24–26]. Applications to membrane proteins have been limited to the design of single-pass TMH associations[27–29].

Here, we developed a computational approach for modeling and designing quaternary structures of multi-pass membrane receptors. Using this method and given that CXCR4 form homodimers that can be regulated by ligands[10,11], we engineered the CXCR4 to associate into distinct oligomeric structures that recruited and activated intracellular signaling proteins differently. Altogether, our study reveals that quaternary structures constitute important unforeseen structural determinants of GPCR biased signaling and identified a common conformational switch at the dimer interface of several GPCRs that differentially control β-arrestin engagement versus G protein signaling. The approach is general and should prove useful for reprogramming cellular functions through designed receptor associations.

## Results

### Computational approach for modeling and designing multi-pass receptor oligomers

We developed an approach to model and design multi-pass membrane protein associations with precise quaternary structures, stabilities, and signaling functions (Fig. 1, Supplementary Fig. 1). We call the method QUESTS which stands for QUaternary rEceptor STate design for Signaling selectivity. The method builds GPCR monomeric structures in distinct active and inactive states, docks them to identify possible modes of protomer associations into homodimers, and designs the binding interfaces to generate quaternary structures with distinct dimer stabilities, conformations, and propensity to recruit and activate specific intracellular signaling proteins. In this study, an active state model refers to a GPCR in an active state conformation modeled using a receptor structure bound to an agonist and G protein or an agonist and β-arrestin as templates.

We applied QUESTS to reprogram the homo-dimeric structure and function of CXCR4, a GPCR from the chemokine receptor family. We chose CXCR4 because it is a critical signaling hub involved in immune responses[7,30] and HIV infection, as well as a receptor for which multiple experimental lines of evidence supporting the formation of constitutive homo-oligomers and its regulation by ligands exists[10,11].

We first modeled CXCR4 WT monomers in inactive and active states (Fig. 1, Supplementary Fig. 1). For instance, the active state model of CXCR4 was obtained from the active state structure of the homologous viral GPCR US28 (PDB 4XT1) using the method IPHoLD which integrates homology modeling and ligand docking[31]. The CXCR4 WT monomers in the inactive state were taken from the antagonist-bound CXCR4 WT structure (PDB 3ODU) after energy minimization of the X-ray coordinates. The CXCR4 WT monomers were assembled into inactive or active state dimers along different dimer binding interfaces involving TMHs 4, 5, and 6. We found that, in both the inactive and active states, the dimer WT models populated primarily an open-dimer

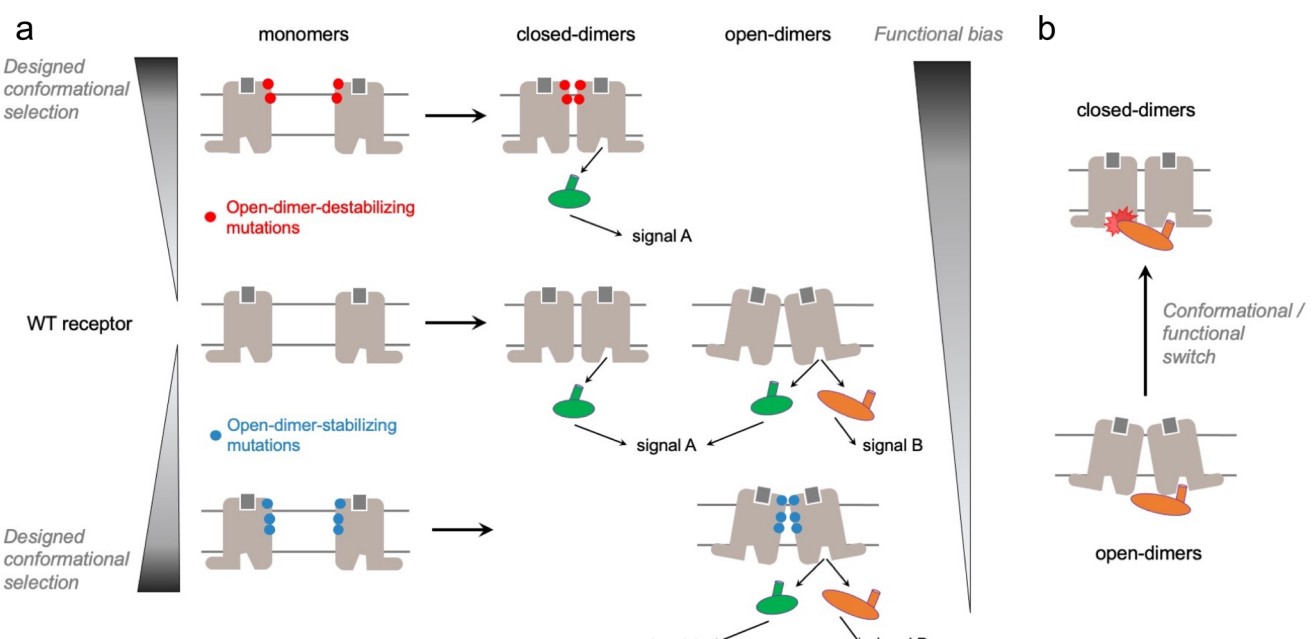

**Fig. 1 | Computational modeling and design of GPCR associations with reprogrammed structures and functions using QUESTS. a** Framework for the modeling and design of specific receptor quaternary active state conformations eliciting various degree of functional selectivity. The WT receptor modeled in the active state is assembled into dimers and then into ternary complex with G proteins (green) or β-arrestin (orange) to identify the distribution of quaternary conformations and their ability to recruit intracellular signaling proteins. The dimer

binding interface is redesigned to stabilize and/or destabilize specific quaternary conformations. This design strategy enhances the quaternary conformational selectivity of the receptor and reprograms the functional bias of the receptor oligomer (Supplementary Fig. 1, Methods). **b** Quaternary structural changes act as a functional switch as the closed-dimer conformation interferes with the binding of a GPCR monomer to β-arrestin.

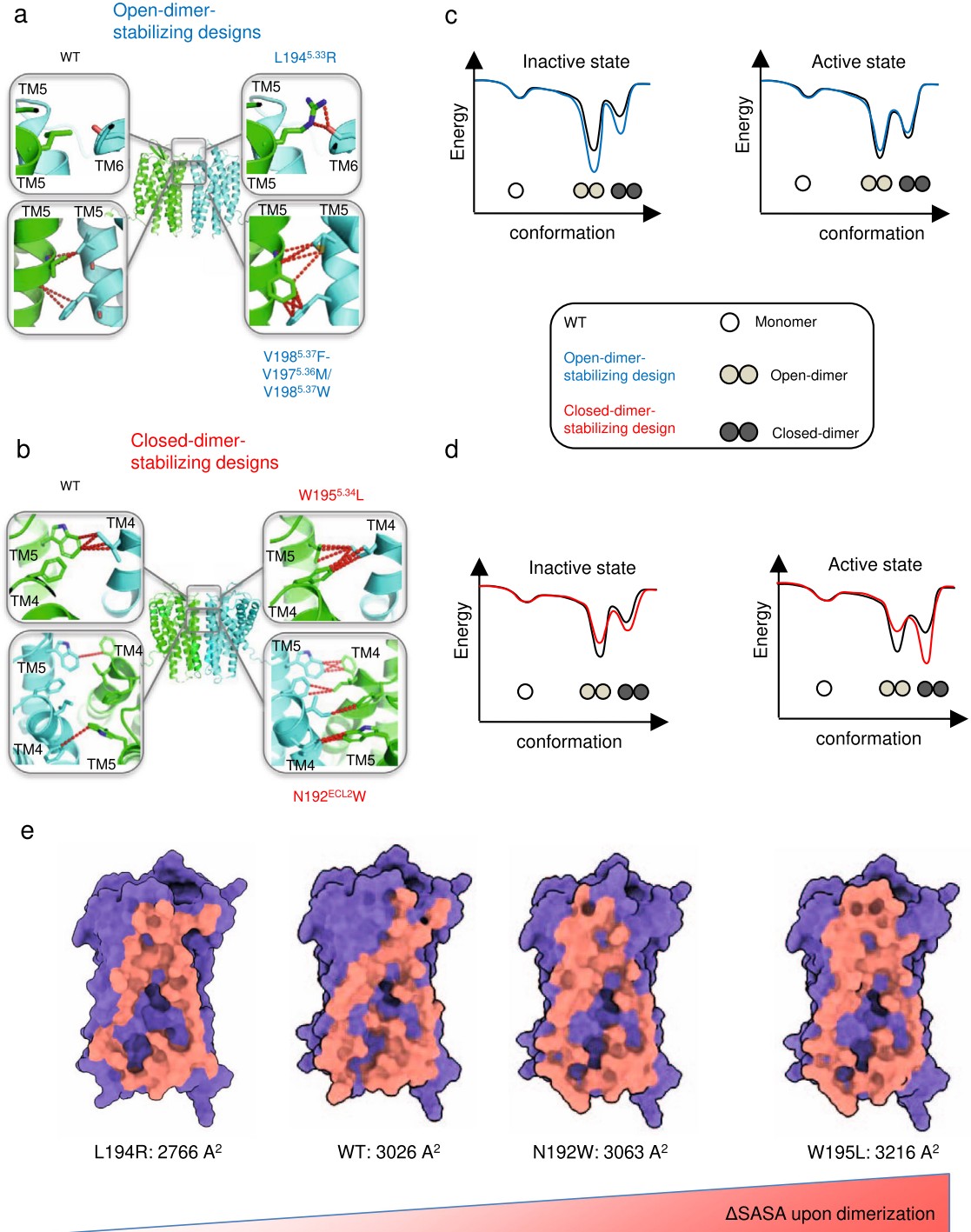

**Fig. 2 | Computational design of CXCR4 associations with specific conformations. a** Mutations designed to selectively stabilize the CXCR4 open-dimer conformation without affecting CXCR4 monomer stability were identified in the extracellular and TMH regions. **b** Mutations designed to selectively stabilize the CXCR4 closed-dimer conformation without affecting CXCR4 monomer stability were identified in the extracellular region. Key atomic contacts are represented as red dotted lines. **c** Schematic conformational energy landscapes of CXCR4 dimerization in the inactive and active states for the open-dimer stabilizing designs. **d** Schematic conformational energy landscapes of CXCR4 dimerization in the inactive and active states for the closed-dimer stabilizing designs. **c**, **d** The dimerization energies reported in Supplementary Table 1 were used to plot the energy landscapes. The monomer energies and energy barriers between states are fictitious and were not predicted by our simulations. **e** Ranking of the CXCR4 variants based on changes in buried surface area (ΔSASA) calculated from the predicted models in the active state. The ΔSASA is reported for the most occupied dimer conformation for each variant: L194R open-dimer state, WT open-dimer state, N192W closed-dimer state, W195L closed-dimer state. Larger buried ΔSASA are predicted to correlate with enhanced dimerization propensity (see Supplementary Table 2).

conformation similar to that observed in the antagonist-bound receptor X-ray structure but also, to a lesser extent, a distinct closed-dimer conformation (Fig. 2, Supplementary Table 1, Supplementary Fig. 2). The distribution between dimer conformations can be deduced

from the difference in binding energy (strength of association) at the distinct dimer interfaces (Supplementary Table 1). Interestingly, while the major open-dimer conformation remains very similar in both signaling states, the minor closed form differs by a slight rotation around

TMH5 between the inactive and active state conformations of the receptor (Supplementary Fig. 2).

To elucidate the function of these different quaternary structures, we then designed TMH and loop binding surfaces to selectively stabilize either the open-dimer or the closed-dimer conformation of the inactive and active state dimer models. QUESTS first searches for combinations of mutations and conformations that modulate the intermolecular interactions between the monomers without affecting the monomer's intrinsic conformational stability and functions. Any design that modifies the dimer binding energies as intended but significantly affect monomer stability is systematically discarded (see Methods). After each round of design, the CXCR4 monomers are assembled into dimers to predict the effects of the designed sequence-structure features on the distribution of quaternary structures in distinct signaling states. Lastly, the G protein Gi and β-arrestin are docked and assembled onto the designed CXCR4 active state dimers to predict whether the engineered receptors would effectively engage and activate these intracellular signaling proteins. The cycles of design, flexible docking and ternary complex assembly are repeated until the calculations converge to significant predicted reprogramming of the quaternary structure and functional selectivity of the designed CXCR4 oligomers (Fig. 1, Supplementary Fig. 1).

## Designing CXCR4 dimers with selective conformations and intracellular functions

From our in silico design screen, we first selected three engineered CXCR4s predicted to dimerize with greater propensity than CXCR4 WT in the open-dimer conformation (Fig. 2a, c, Supplementary Table 1). The designs involved key conformational lock motifs stabilizing the open-dimer conformation (Fig. 2a, c). The L194$^{5.33}$K and the L194$^{5.33}$R design introduced a set of strong and conformationally selective polar contacts between the extracellular sides of TMH5s of two protomers predicted to stabilize the dimer interface by 2.4 Rosetta Energy Units (REU) (Fig. 2a, Supplementary Table 1). The triplet design, formed by the V198$^{5.37}$F-V197$^{5.36}$M mutation on one protomer and the V198$^{5.37}$W on the other protomer, encoded a new network of optimal hydrophobic contacts bridging the membrane-embedded core of the dimer-binding interface between TMH5s (Fig. 2a). When modeled in the active state, these designs primarily dimerized in an open conformation that could readily form tight active state complex structures with both Gi and β-arrestin (Fig. 1, Supplementary Table 1, Supplementary Fig. 3).

Conversely, we also engineered two binding surfaces predicted to stabilize the closed-dimer conformation (Fig. 2b, d, Supplementary Table 1). We selected these "closed-dimer-stabilizing" designs, because, unlike WT, they preferentially assemble into closed-dimer conformations that form tight active-state complex structures with Gi but not with β-arrestin (Fig. 2b, d, Supplementary Fig. 3). We found that steric hindrance prevents the optimal interaction of β-arrestin's finger loop in the intracellular binding groove of CXCR4 when the receptor occupies the closed-dimer conformation. Specifically, our models predict that regions of close contacts between the β-arrestin and the open-dimer CXCR4 (i.e. helix 8 of CXCR4 monomer 2 with the C-tip of β-arrestin, and ICL2 of CXCR4 monomer 1 with the C-loop of β-arrestin) would be disrupted in the closed-dimer conformation (Supplementary Fig. 4). Both designed interfaces (that we name W195$^{5.34}$L and N192$^{ECL2}$W design switches) involved distinct conformational switch motifs stabilizing the closed-dimer conformation, especially when the receptor occupies the signaling active state (Fig. 2b, Supplementary Table 1). The W195$^{5.34}$L design switch increased the packing of TMH4 and 5 across the extracellular side of the binding interface, stabilizing the closed-dimer conformation through additional van der Waals contacts (Fig. 2b). The N192$^{ECL2}$W design switch induced several conformational changes in a neighboring layer of residues buried at the

dimer interface, creating new key hydrophobic interactions stabilizing the closed form (Fig. 2b).

By simulating the association for the WT and the designed CXCR4 monomers, we identified important differences in the stability of the dimer conformations and hence in the distribution of the monomer and dimer forms in the inactive and active states of the receptor (Fig. 2b, d, Supplementary Tables 1, 2). Although QUESTS does not rigorously calculate free energies of dimerization, we could derive an apparent dimerization propensity score relative to WT for the different CXCR4 variants (Methods, Supplementary Table 2). The significant difference in dimerization propensities between the designs described below stemmed directly from the distinct calculated stabilities of the dimer conformations (Methods, Supplementary Table 1). In the inactive state, the "closed-dimer-stabilizing" N192$^{ECL2}$W and W195$^{5.34}$L designs formed weaker dimers while the "open-dimer-stabilizing" L194$^{5.33}$R design formed stronger dimers than WT, suggesting that the "closed-dimer-stabilizing" designs would occupy more often the monomeric form in the inactive state. The reverse scenario was observed in the active state. While the dimerization propensity of the L194$^{5.33}$R design was lower than WT, the W195$^{5.34}$L design formed the most stable active state dimers among all variants. Interestingly, the dimerization propensities were consistent with the changes in receptor buried surface areas upon self-association (Fig. 2e), except for the L194$^{5.33}$R design which stabilizes the dimer interface through strong polar interactions instead of VDW contacts (Fig. 2a).

Our calculations suggested also important differences in the distribution between dimer conformations (Fig. 2b, d, Supplementary Table 1). Concerning the WT receptor, we observed that the closed-dimer conformation was significantly more stable in the active state, indicating a relative shift toward the closed form in that state. By contrast, virtually no difference in the closed-dimer conformation stability between the inactive and active states was observed for the "open-dimer-stabilizing" designs (L194$^{5.33}$R switch). The largest changes in dimer populations between inactive and active signaling states were observed for the "closed-dimer-stabilizing" designs (W195$^{5.34}$L switch). Despite a significant stabilization of the closed-dimer conformation, the open dimer remained the most stable form in the inactive state and the W195$^{5.34}$L variant still predominantly populated the open-dimer structure in that state. However, the distribution between open and closed conformation of the W195$^{5.34}$L variant was reversed in the active state and the closed form became the most stable and dominant structure. Overall, the W195$^{5.34}$L design was found to be most stable in the active state closed-dimer form (Supplementary Table 1).

## Designed CXCR4 receptors dimerize in distinct conformations

We validated the predicted designed oligomeric CXCR4 structures and functions using an ensemble of cell-based experiments.

We first measured constitutive and CXCL12 agonist-promoted CXCR4 dimerization in living HEK293T cells by BRET using CXCR4-RLuc and CXCR4-YFP constructs (Fig. 3a). A large constitutive BRET signal was observed for the WT receptor which, as previously reported[10,11], further increased upon activation by agonist (Fig. 3a, Supplementary Fig. 5, Supplementary Fig. 6 for CXCR4 cell surface expression levels). This increase in BRET can be interpreted as a change in conformation within dimers and a shift toward the closed-dimer form or as an increase in dimer population upon activation that were both suggested by our calculations (Supplementary Table 1). Although both phenomena most likely contribute to the increase, their relative contribution cannot be determined from the BRET data or, to our knowledge, any other experimental approach. Consistent with the "open-dimer-stabilizing" designs associating in a similar open conformation than WT, the constitutive BRET signals measured for these designs (L194$^{5.33}$K/L194$^{5.33}$K, L194$^{5.33}$R/L194$^{5.33}$R,

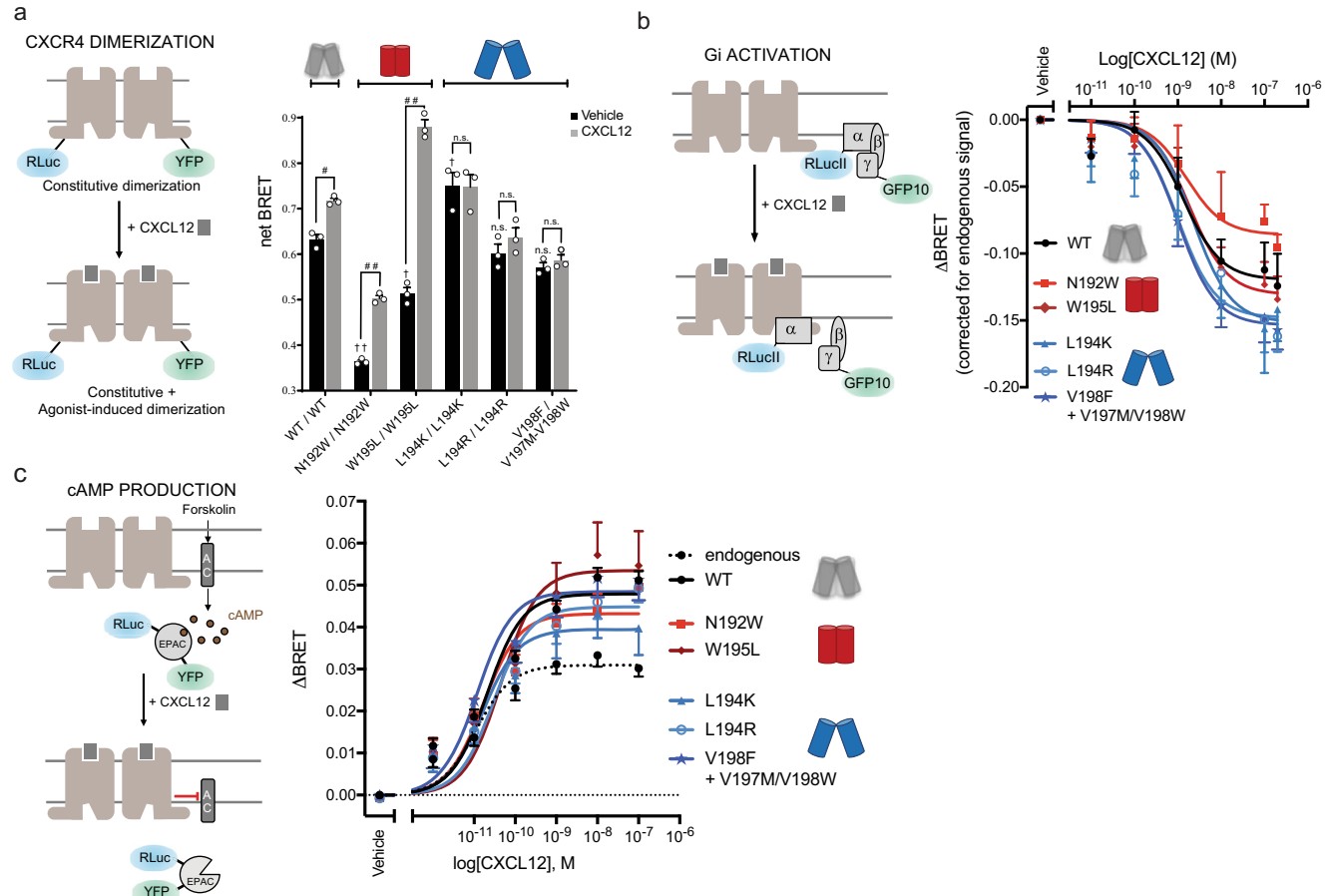

**Fig. 3 | CXCR4 association and Gi activation. a** (Left) Schematic representation of the CXCR4 dimerization BRET-based assay. (Right) CXCR4 association was measured by BRET before (black) and after agonist stimulation (gray) in HEK293T cells transfected with CXCR4-RLuc and its counterpart CXCR4-YFP, WT, or mutant as indicated. BRET$_{480-YFP}$ was measured after the addition of coel-h (10 min) and CXCL12 (15 min). Data shown represent the mean ± SEM of three independent experiments and are expressed as net BRET (calculated by subtracting background luminescence). Statistical significance was assessed using a two-way ANOVA followed by a Šídák's multiple comparisons test: $^{\#}p = 0.007$, $^{\#\#}p < 0.0001$, n.s. not significant $p > 0.05$ are used to compare BRET values between basal to CXCL12-treated conditions and $^{\dagger}p = 0.0004$, $^{\dagger\dagger}p < 0.0001$ are used to compare basal BRET values between the mutants. **b** (Left) Schematic representation of the BRET-based ligand-induced Gi activation assay. (Right) CXCL12-promoted Gi activation

measured by BRET in HEK293T cells transfected with HA-CXCR4, WT or mutant as indicated, Gαi1-RLucII, Gβ1, and Gγ2-GFP10. BRET$_{400-GFP10}$ was measured after the addition of coel-400a (10 min) and CXCL12 (3 min). **c** (Left) Schematic representation of the BRET-based EPAC sensor to measure cAMP production. (Right) CXCL12-promoted EPAC inhibition was measured by BRET in HEK293T cells transfected with HA-CXCR4, WT or mutant as indicated, and RLuc-EPAC-YFP. BRET$_{480-YFP}$, reporting the conformation rearrangement of the EPAC sensor from an open to a closed conformation, was measured after the addition of coel-h (10 min) and CXCL12 (5 min). **b**, **c** CXCR4 mutations predicted to stabilize the open-dimer or the closed-dimer conformation are annotated with a blue or red dimer symbol, respectively. Data shown represent the mean ± SEM of at least three independent experiments and are expressed as ΔBRET (agonist-promoted BRET).

V198$^{5.37}$F/V197$^{5.36}$M-V198$^{5.37}$W), were either similar to or slightly larger than WT. However, unlike what is seen for the WT receptor, we did not observe any significant BRET increase upon agonist stimulation. These results suggest that stabilization of the open-dimer conformation prevents further agonist-induced conformational changes across the binding interface and locks the receptor dimer in a constitutive open-dimer conformation, consistent with the lack of stabilization of the closed-dimer form upon receptor activation in our simulations (Supplementary Table 1). In the specific case of the L194$^{5.33}$K design, the lack of BRET increase upon stimulation could also result in part from the designed receptors occupying more frequently the dimer state than the WT receptor, even without stimulus, as suggested by the significantly increased constitutive BRET signals measured for that design.

The BRET signals measured for the "closed-dimer-stabilizing" designs (N192$^{ECL2}$W/N192$^{ECL2}$W, W195$^{5.34}$L/W195$^{5.34}$L) without ligand stimulus were significantly lower than WT. These observations are consistent with the designs still predominantly occupying the open conformation in the inactive state (Supplementary Table 1) and

forming overall weaker dimers than WT (Supplementary Table 2) that may result in a greater proportion of receptor in the monomeric state. Upon agonist stimulation, however, we observed a larger increase in net BRET signal compared to WT, especially for W195$^{5.34}$L in agreement with the large predicted changes in dimer conformation favoring the closed-state and increased dimerization propensity upon receptor activation (Supplementary Tables 1, 2). These results suggest that the "closed-dimer-stabilizing" receptors constitutively dimerize less than WT and display stronger propensity to associate in the closed-dimer form upon agonist stimulation.

Overall, we observed a consistent trend between predicted closed-dimer stabilization and increase in Delta BRET upon activation (Supplementary Fig. 7). These results suggest that major conformational changes and population shifts towards the closed-dimer form can readily occur in the active state when triggered by strong switching mutations such as W195$^{5.34}$L. Concerning the effects of the designs on dimerization, except for L194$^{5.33}$R, we observed a qualitative trend between the calculated propensities and BRET measurements (Supplementary Table 2), which suggest that designed structural

interactions may impact the dimerization propensity as predicted by the design calculations.

In summary, the BRET measurements validate the designed CXCR4-dimer structures and indicate that receptor dimers with distinct strengths of associations and quaternary conformations can be rationally engineered using our approach.

## Designed CXCR4 receptors activate distinct intracellular signaling proteins

According to our calculations, the two classes of designed receptors should display distinct propensity to bind and activate intracellular signaling proteins. While the receptors dimerizing in the open conformation should recruit both Gi and β-arrestin, the receptors preferentially dimerizing in a closed conformation should couple strongly to Gi only.

To validate these predictions, we measured Gi activation and β-arrestin recruitment to CXCR4 using BRET-based assays in HEK293 cells. Consistent with the active state modeling, both classes of designed CXCR4 dimers were able to activate Gi similarly to WT, as measured by the agonist-induced dissociation of the heterotrimeric Gi protein subunits (Fig. 3b) and the inhibition of cAMP production (Fig. 3c). As shown in Supplementary Fig. 6d, HEK293 cells endogenously express a low level of CXCR4 that result in a background CXCL12-promoted cAMP inhibition that can easily be distinguished from the signal generated by the transfected WT or mutant receptors (Fig. 3c). No such background signal is observed in the BRET-based Gi activation or β-arrestin recruitment assays due to the lower level of amplification of these assays. Both assays clearly indicated that the mutations did not affect the ability of the receptor to activate Gi.

β-arrestin recruitment was measured using BRET reporting directly the interaction between CXCR4-RLuc and β-arrestin-2-YFP (in HEK293 cells)[32] or ebBRET[33] monitoring the interaction between β-arrestin-2-RLuc and the lipid-modified rGFP-CAAX anchored at the cell membrane. Both assays consistently showed that the "open-dimer-stabilizing" designs recruited β-arrestin very effectively and similarly to WT upon agonist stimulus (Fig. 4a, Supplementary Fig. 8). On the contrary, and in agreement with our predictions, the "closed-dimer-stabilizing" designs had largely impaired β-arrestin recruitment abilities. Specifically, while β-arrestin-2 coupling to the N192$^{ECL2}$W design was considerably reduced compared to WT, virtually no recruitment signals could be measured for the W195$^{5.34}$L design (Fig. 4a, Supplementary Fig. 8). The differences in β-arrestin recruitment were not due to difference in the expression levels of the different mutants as they showed similar cell surface expression as assessed by ELISA (Supplementary Fig. 6). Given that phosphorylation of GPCRs is known to increase the affinity of β-arrestin for the active forms of receptors, we assessed the impact of W195$^{5.34}$L on the agonist-promoted phosphorylation of CXCR4. As shown in Fig. 4b, despite a small reduction in the basal phosphorylation level, the same CXCL12-promoted phosphorylation was observed for WT- and W195$^{5.34}$L-CXCR4, indicating that the loss of agonist-promoted β-arrestin recruitment is not due to a phosphorylation defect.

Consistent with the previously reported role of β-arrestin in ERK activation[34], the W195$^{5.34}$L design showed a reduced level of ERK phosphorylation compared to WT, suggesting that the scaffolding function supported by β-arrestin was affected (Fig. 4c). Because of a high background CXCL12-promoted ERK activity in HEK293 cells, the ERK assay was performed in U87.GM cells that lack endogenous CXCR4 in which WT- and W195$^{5.34}$L-CXCR4 were heterologously expressed at equivalent expression levels (Supplementary Fig. 6e). These data show that the designed change in dimerization resulted into a functional outcome at the signaling level reflected by a blunted ERK response, a result that is consistent with the reduced β-arrestin recruitment observed for this mutant.

## New structural mechanism of GPCR-mediated biased signaling

Overall, our designs reveal an unforeseen structural mechanism of GPCR-mediated biased signaling. Molecular determinants of biased signaling identified so far were primarily encoded by specific sequence motifs and conformations of receptor monomers[35]. However, Gi-mediated CXCR4 signaling triggering important functions such as chemotaxis was recently found to depend on the formation of specific receptor nanoclusters at the cell surface[23]. These oligomers are controlled by specific structural motifs on the lipid-exposed intracellular surface of TMH6, that is remote from the dimer interface studied here (Fig. 5a). Mutations of the corresponding residues on TMH6 resulted in nanocluster-defective receptor variants with severely impaired Gi-mediated signaling, suggesting that this CXCR4 oligomerization surface constitutes a Gi bias signaling switch. On the other hand, our study demonstrates that the extracellular dimerization surface primarily constituted by TMH5 residues can control the selective recruitment of the other main class of GPCR signaling and regulating partners, β-arrestin. Since Gi coupling remains insensitive to the precise dimer structure mediated by TMH5 contacts, we propose that this binding surface constitutes a β-arrestin bias signaling switch. This is consistent with our modeling suggesting that the active close-dimer conformations prevent the engagement of the β-arrestin finger loop of the receptor by the cradle core of the receptor through steric hindrance (Supplementary Fig. 4).

Since the structural motifs identified at the surface of CXCR4 monomers control a key signaling pathway conserved in most GPCRs, we wondered whether similar binding surfaces could be identified in other receptors. We first performed a sequence alignment of CXCR4s from various organisms and found that the native residues at the designed dimerization hotspot positions were highly conserved in CXCR4s through evolution, supporting an important functional role for this region of the receptor (Fig. 5b). Strikingly, a similar analysis revealed that these positions are poorly conserved in other human chemokine receptors with the exception of W195$^{5.34}$ (Fig. 5b). Interestingly, while no other chemokine receptors have been crystallized in a dimeric form involving TMH5-mediated contacts, the position of W5.34 in CXCR1 (PDB 2LNL), CCR2 (PDB 5T1A), CCR5 (PDB 4MBS) and Y5.34 in chemokine-related US28 (PDB 4XT1) was found to be superimposable to that in CXCR4 (Fig. 5c). We also found conserved aromatic residues at position 5.34 in P2Y and other peptide-binding receptors which are known to dimerize (Fig. 5b).

Since a single mutation at the extracellular tip of TM5 (W5.34) was sufficient to disrupt β-arrestin recruitment at CXCR4, we wondered whether that particular position could also constitute a β-arrestin bias signaling switch in other class A GPCRs. To validate this hypothesis, we identified two additional GPCRs from the peptide-binding receptor subfamily that are known to dimerize, the mu-opioid receptor (μOR) and the type-2 vasopressin receptor (V2R), that bear a tryptophan at position 5.34 or 5.33, respectively (Supplementary Fig. 9). We investigated the effect of mutating the native tryptophan at these positions to alanine to assess its role on the receptor signaling functions. Receptor signaling was measured using distinct BRET sensors monitoring β-arrestin and Go for μOR, and β-arrestin, and Gs for V2R (Fig. 6). While signaling through the G proteins was moderately affected by the W5.34/33A mutation in both receptors (V2R: 85% and μOR: 61% of WT efficacy, Fig. 6), μOR and V2R receptor mutants were more strongly impaired in β-arrestin signaling (<25% of WT efficacy, Fig. 6). These results indicate that the aromatic residue at position 5.34/33 preferentially controls the signaling efficacy through the β-arrestin pathway and plays the role of a signaling switch in receptors that belong to distinct peptide-binding GPCR families.

To better understand the structure-function underpinnings of the mutational effects and assess whether a common structural mechanism underlies the function of the identified switches in the studied

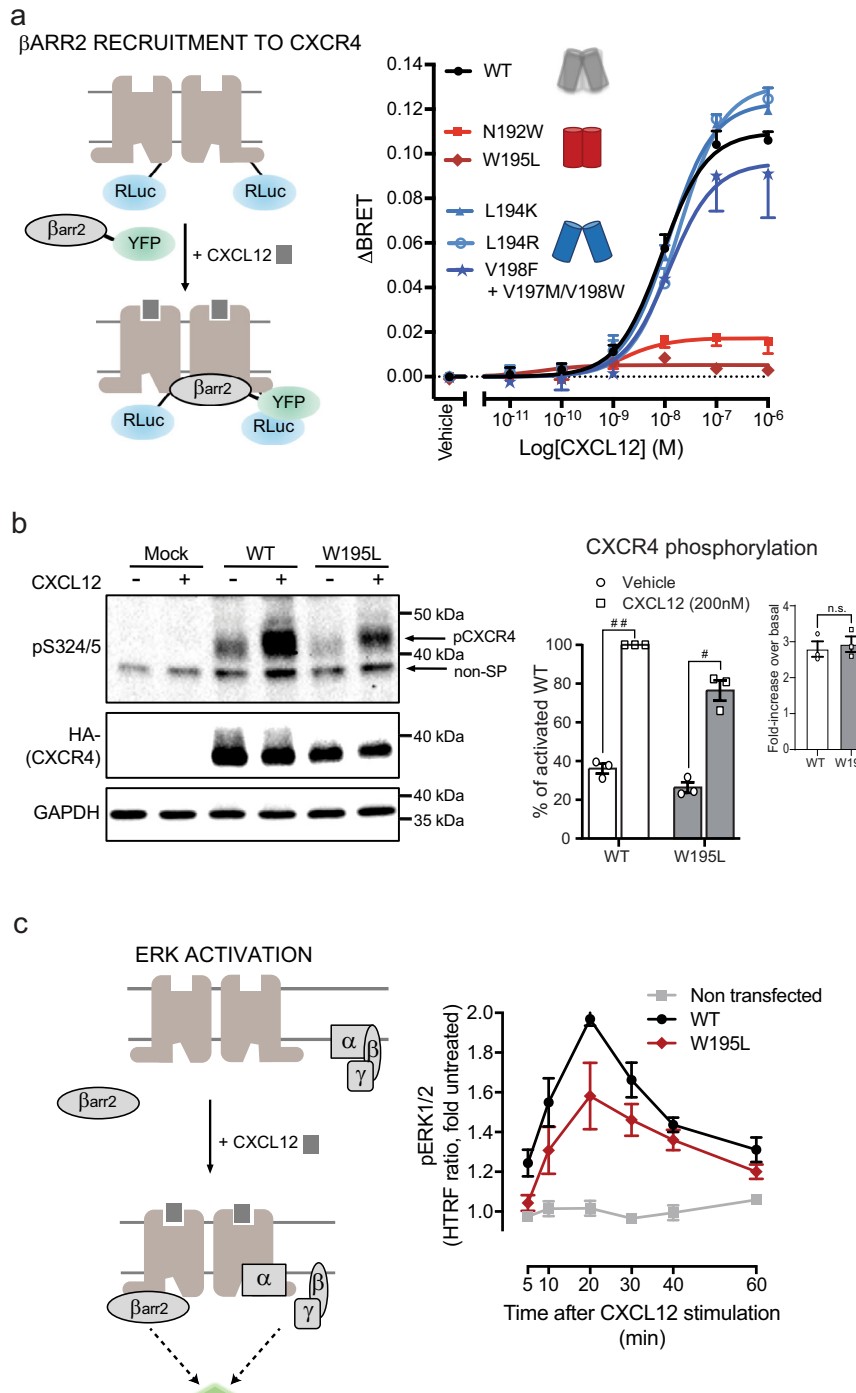

**Fig. 4 | β-arrestin recruitment and ERK activation. a** (Left) Schematic representation of the BRET-based ligand-induced β-arrestin-2 (βarr2) translocation assay. (Right) CXCL12-promoted βarr2 recruitment to CXCR4 measured by BRET in HEK293T cells transfected with CXCR4-RLuc, WT or mutant as indicated, and βarr2-YFP. $BRET_{480-YFP}$ between CXCR4-RLuc and βarr2-YFP was measured after the addition of coel-h (10 min) and CXCL12 (15 min). Data shown represent the mean ± SEM of at least three independent experiments and are represented as ΔBRET. **b** (Left) Phosphorylation at S324/5 of WT and W195[5.34]L CXCR4 promoted by stimulation with 200 nM CXCL12 for 30 min detected using an anti pS324/5 antibody (pCXCR4 indicates the CXCR4-S324/5 phosphorylation band; non-SP correspond to a non-specific band). (Right) Quantification of phosphorylation

bands normalized as a function of the intensity of the total HA-CXCR4 detected using an anti-HA antibody. Shown in the inset is the fold increase in phosphorylation over basal levels. Data shown represent the mean ± SEM of three independent experiments. Statistical significance was assessed using unpaired $t$ test. $^{\#}p = 0.001$, $^{\#\#}p < 0.0001$, n.s. not significant $p > 0.05$. **c** (Left) Schematic representation of ERK activation by CXCR4. (Right) ERK phosphorylation in U87 stably expressing equivalent levels of WT and W195[5.34]L CXCR4 induced by stimulation with 10 nM CXCL12 for the indicated times was monitored by HTRF. CXCR4 mutations predicted to stabilize the open-dimer or the closed-dimer conformation are annotated with a blue or red dimer symbol, respectively. Data shown represent the mean ± SEM of at least three independent experiments.

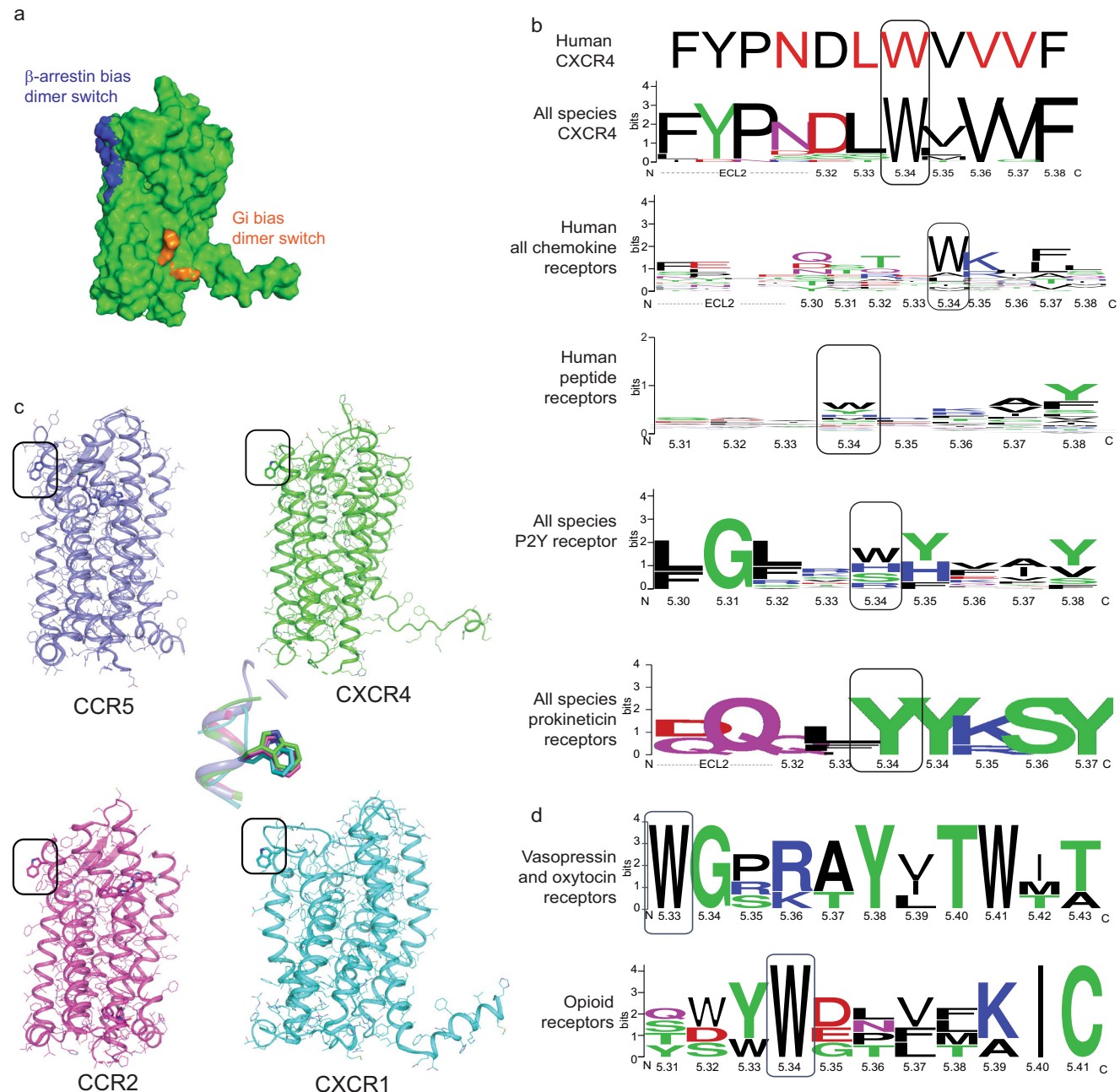

**Fig. 5 | Distinct quaternary structures selectively control G protein and β-arrestin recruitment. a** Surface representation of the CXCR4 inactive state monomeric structure highlighting the distinct oligomerization interfaces controlling either β-arrestin recruitment (extracellular side and TM core of TMH5, blue) or Gi activation and nanocluster formation[23] (intracellular side of TMH6, orange). **b** The hotspot binding sites controlling CXCR4 oligomerization through TMH5 (designed residues in red) are poorly conserved in the chemokine receptor family, except for the β-arrestin signaling switch W5.34. Aromatic residues are highly enriched at position 5.34 of other dimerizing GPCR families. **c** Conserved position and conformation of W5.34 in human chemokine receptor X-ray structures. The superposition of W5.34 conformations is shown in the center. **d** Conservation of the β-arrestin signaling switch in the TM5 of vasopressin/oxytocin receptors and opioid receptors.

receptors, we investigated the structural impact of the tryptophan to alanine substitution. We focused our analysis on µOR, as a broad range of structural and functional evidence indicate that this receptor strongly homodimerizes in cell membranes[36,37]. In particular, a high-resolution structure of the murine µOR in the inactive state revealed a homodimer stabilized by an extensive binding interface between TMH5 and TMH6. Using our computational quaternary structure modeling approach, we modeled WT and W5.34 A µOR homodimers in active signaling complexes bound to either G protein Go or β-arrestin. Our simulations revealed that µOR in the active signaling state mainly

adopts a major "open" and a minor "wide-open" homodimer conformational state (Supplementary Fig. 10). The open-dimer form of µOR was found to strongly bind to β-arrestin, while the wide-open dimer interacted considerably less well with that protein (Supplementary Figs. 11, 12, Supplementary Table 3). By contrast, both homodimer conformations were able to strongly recruit Go (Supplementary Fig. 11, Supplementary Table 3). W5.34 was found at the dimer interface of all µOR homodimers but involved in different sets of interactions. Consequently, the W5.34 A mutation displayed distinct effects on the dimer structures, destabilizing the major open dimer

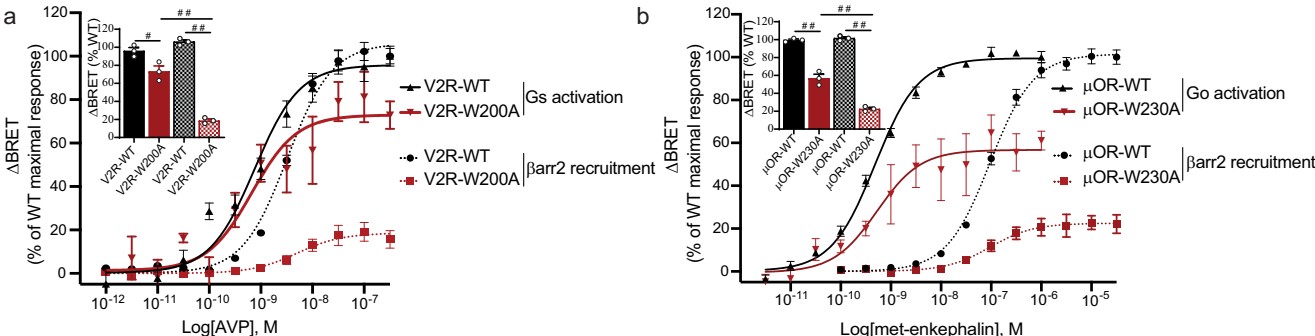

**Fig. 6 | W5.34/33 is a common biased signaling switch in dimerizing peptide-binding GPCRs. a**, **b** Ligand-promoted Gs (V2R) or Go (μOR) and βarr2 recruitment to the membrane in the presence of V2R-WT or V2R-W200$^{5.33}$A (**a**), and μOR-WT or μOR-W230$^{5.34}$A (**b**). Gs and Go activation were detected by monitoring the dissociation between Gα and Gβγ by BRET whereas β-arrestin recruitment to the plasma membrane was assessed in cells transfected with rGFP-CAAX, WT, or mutant receptors, as indicated, and βarr2-RlucII. Data shown represent the mean ± SEM of three independent experiments and are represented as ΔBRET normalized to the maximal response of the WT receptor. The insets represent the maximal agonist-promoted responses for G protein activation (solid bars) and β-arrestin-2 recruitment checkered bars. Statistical significance was assessed using a one-way ANOVA followed by Tukey's multiple comparison test: $^{\#}p = 0.012$, $^{\#\#}p < 0.0001$.

and stabilizing the minor wide-open form (Supplementary Table 3). The simulations corroborate the experimental observations and provide a structural explanation as to why W5.34 A preferentially decreases signaling through β-arrestin. Since no high-resolution structure of a dimer of V2R is available, similar simulations were not attempted on that receptor. Nevertheless, from the available agonist-bound monomeric V2R structure bound to Gs (PDB 7DW9) it is evident that W5.33 in V2R points in a very similar direction than W5.34 in CXCR4 and would be involved in a similar dimer interface than CXCR4 and μOR (Supplementary Fig. 9).

To further support our structural interpretations of the observed biased signaling, we performed structural predictions of the studied complexes using Alpha-Fold multimer (AF2)[38,39]. While AF2 was not able to predict the CXCR4-dimer conformation on its own, it did recapitulate our predicted conformations or the X-ray structure when these structures were given as template (Supplementary Fig. 13). AF2 predictions of μOR dimers converged also to our modeled conformations when provided a template. Concerning the G protein and β-arrestin-bound complex structures, we obtained good agreement between AF2 and our modeling approach. Even without template, AF2 recapitulated our β-arrestin-2 bound CXCR4 models. In fact, all AF2 models predicted the orientation of the β-arrestin-2 and the main interactions between the arrestin finger loop and the receptor observed in our model (Supplementary Fig. 14). Importantly, AF2 predicted the same steric hindrance between arrestin and the receptor in the "closed-state" dimer that we identified using our approach (Supplementary Fig. 15). Lastly, while the AF2 predicted Gi-orientation in the Gi-bound CXCR4 complex was slightly different than in our models, the bound conformation of the G protein and interactions with the receptor were not sensitive to the dimer conformation (Supplementary Fig. 15), consistent with our modeling calculations. Overall, the AF2 models support the quaternary structure-based mechanism of biased signaling that we uncovered and report in this study.

Overall, our findings imply that residues at position 5.34/33 control β-arrestin signaling of CXCR4, μOR, and V2R by acting as a bias switch at quaternary interfaces. While the 3 receptors are functionally unrelated, belong to 3 distinct receptor subfamilies, and couple to different G proteins, this specific mechanism may not universally apply to all GPCRs. We found 6 class A GPCR structures in the pdb that form an extensive and symmetric homodimer interface. Analysis of these structures suggests that receptors mainly self-associate through either TMHs 4, 5, and 6 (e.g., in CXCR4, μOR) or TMHs 1,2,7 and 8 (e.g. in rhodopsin, beta

1 adrenergic receptors), implying that the latter could use other functional selectivity switches to regulate distinct signaling properties. Nevertheless, our study provides solid evidence from 3 unrelated receptors belonging to one of the main class of structural dimers that functional selectivity switches can exist at a specific transmembrane helical dimer interface thus defining a new molecular mechanism of regulating GPCR signaling.

## Discussion

Membrane protein oligomers are ubiquitously observed in cell membranes and have been widely investigated using structural, spectroscopic, and mutagenesis approaches[40]. However, how specific self-associations and quaternary structures control selective protein functions has remained elusive for many classes of multi-pass membrane proteins, including GPCRs. We developed QUESTS, a general computational modeling, and design approach that enables the precise design of binding surfaces and interactions to perturb native or create novel receptor oligomeric structures and associated functions.

A large fraction of GPCRs can activate multiple signaling pathways. This promiscuity has proven a challenge for the development of selective therapeutics since drugs targeting the canonical extracellular ligand binding site of GPCRs often trigger several intracellular functions, beyond the therapeutically relevant one(s) leading to undesirable side-effects[41]. In this study, we uncovered and engineered hotspot dimerization conformational switches on the extracellular side of CXCR4 and μOR that controlled the precise receptor dimeric structure and the selective activation of intracellular signaling pathways. Interestingly, we also identified a biased signaling hotspot at the same location in another strongly homodimerizing peptide-binding GPCR, V2R, but for which a high-resolution dimer structure has not yet been determined. Altogether, the results suggest that specific positions at dimer interfaces can act as conformational switches to control biased signaling in GPCRs.

The extracellular locations of these biased signaling switches suggest that the sites are druggable. The signaling regulatory mechanism controlled by specific receptor oligomeric structures emerging from our study opens new avenues for selective pharmacological treatments that do not perturb receptor monomeric structures and associated signaling functions.

Overall, our approach should prove useful for designing multi-pass membrane protein associations with novel structures and functions, and expand protein design toolkits for engineered cell-based therapies and synthetic biology applications.

## Methods

### Modeling CXCR4 inactive state monomer and homodimer structures

The X-ray structure of the antagonist-bound human chemokine receptor CXCR4 homodimer (PDB 3ODU) served as a starting template for modeling the CXCR4 inactive state monomer. After removal of detergent and lipid molecules, the two receptor molecules were separated from the dimer structure and the region corresponding to the binding interface was relaxed in implicit lipid membrane environment (The RMSD between the relaxed structure and the starting antagonist-bound X-ray structure was 0.1 Å over Cα atoms). The lowest energy relaxed CXCR4 monomer structure was selected as a representative model of the CXCR4 inactive state monomer.

The symmetric flexible docking mode of RosettaMembrane[28] involving inter-monomer rigid-body movements and intra-monomer conformational flexibility was then applied to model CXCR4 homodimer inactive state structures. 10,000 homodimer models were generated starting from the selected CXCR4 inactive state monomer model. The 10% lowest homodimer interface energy CXCR4 homodimer models were selected and then filtered by inter-protomer angles to select quaternary structures that had both optimal homodimer binding energies and proper membrane insertion. Specifically, the relative orientation of the monomers in the X-ray homodimer structure is characterized by an interhelical angle between helix 5 of 52 degrees which ensures optimal membrane embedding. Hence, all models where such angle was no larger than 85 degrees and no less than −50 degrees were considered compatible with proper embedding. Overall, 80% of the models selected by interface energy were kept after applying this relative orientation filter.

These homodimer models were then clustered by dimer-specific geometric parameters across the dimer binding interface (i.e., θ and d, as described in Supplementary Fig. 2) for major dimer orientation analysis. We used the hdbscan-clustering method, which is a density-based clustering method based on hierarchical density estimates[42]. A majority of the models clustered in two large families of distinct dimer conformations (i.e., closed or open) characterized by very different interhelical angles and distances between TMH5 as described in Supplementary Fig. 2. The lowest-energy structure from each cluster was selected as the representative model of each specific (i.e., closed or open) homodimer conformation.

### Quaternary structure assembly of CXCR4 active state dimer complexes bound to G protein or β-arrestin

The general strategy for modeling G protein or β-arrestin-bound CXCR4 active state homodimers involved the following steps: First, the CXCR4 monomer was modeled in the active state conformation and then assembled into homodimers. Lastly, the G protein Gi and β-arrestin-2 were also modeled and assembled onto the CXCR4 active state dimers to generate an optimal signaling complex. The same procedure was applied to model the WT and designed CXCR4 quaternary structures.

**Modeling CXCR4 active state monomer structures.** We applied RosettaMembrane homology modeling method[31,43] to model the agonist-bound conformations of a CXCR4 active state monomer. We used the nanobody and chemokine-bound active state viral GPCR (PDB 4XT1, Sequence identity = 30%) as a template because it displayed the highest sequence homology to CXCR4 among active state GPCR structures. 50,000 models of CXCR4 monomer were generated and the 10% lowest-energy models were clustered based on Cα RMSD. The cluster centers of the top 10 largest clusters were used to build models of active state CXCR4 homodimer.

**Modeling CXCR4 active state homodimer structures.** Active-state CXCR4 homodimer structures were modeled using the same approach than for the inactive state models with the exception that 10 starting active state monomer models were considered. The symmetric flexible docking mode of RosettaMembrane[28] was applied on each monomer model, and, after filtering by interhelical angle, all homodimer models were pooled together prior to the final clustering step. The lowest interface energy decoy from the largest clusters were selected for modeling CXCR4-dimer-β-arrestin-2 or CXCR4-dimer-Gi complexes.

**Modeling GPCR-bound β-arrestin-2 conformations.** Arrestin binding to GPCRs mainly involves 3 loops which undergo significant conformational changes upon receptor binding. Since β-arrestin-2 was never crystallized in complex with a GPCR, to increase the chance of identifying optimal CXCR4-β-arrestin-2 binding modes, we modeled the receptor-bound conformations of β-arrestin-2 by homology to that of the close homolog arrestin-1 bound to Rhodopsin (Sequence identity = 60%, PDB 4ZWJ) using Rosetta homology modeling. 10,000 models were generated, and the lowest 10% energy models were clustered. The lowest-energy models of the largest clusters (containing at least 2% of the population) were used to generate CXCR4-dimer-β-arrestin-2 complex structures.

**Assembling β-arrestin-2-CXCR4-dimer active state complexes.** A total of eight β-arrestin-2 models were selected for optimal docking assembly to each selected active CXCR4 homodimer models. Starting conformations were generated by aligning one subunit of the CXCR4 dimer to Rhodopsin receptor and β-arrestin-2 to visual arrestin in the Rhodopsin-arrestin X-ray structure (PDB 4ZWJ). 5000 models were generated by flexible docking perturbation of the starting structure to optimize the interaction between the different domains of β-arrestin-2 and the intracellular regions of the CXCR4 homodimers. The complexes with the lowest interface energy were selected as representative conformations of β-arrestin-2 bound to one CXCR4 homodimer structure model.

**Assembling Gi-CXCR4-dimer active state complexes.** The α-subunit of the Gi protein (Gαi)-CXCR4-dimer structure was modeled before the first X-ray structure of a GPCR-Gi complex was solved. The GPCR-bound active state conformation of the C-terminal domain of Gi (including the α5 C-terminal helix) was modeled from the Gs structure bound to the β2 adrenergic receptor (β2AR) (PDB 3SN6, Sequence Identity >40%). The C-terminal domain model of Gi was grafted onto the N-terminal domain of the GTPγS bound structure of Gi protein α-subunit (PDB 1GIA) to model the full-length GPCR-bound conformation of Gαi. 10,000 models were generated and the lowest-energy 10% models by total energy were clustered. The lowest-energy decoys in the largest clusters were used as representative active state Gαi to assemble CXCR4-dimer-Gαi complex structures.

The starting position of Gαi for docking onto CXCR4 was generated by aligning Gαi and CXCR4 to the β2 adrenergic receptor and Gs protein α-subunit, respectively in their bound active state structure (PDB 3SN6). 5000 models were generated through perturbation of the starting structures to refine the interaction between the downstream effector and CXCR4 models. The docked structures were filtered by interface energy (lowest 1% effector-interface energy) and clustered. The models with the lowest effector-docking interface energy in the largest clusters were selected as representative conformation for further analysis.

### Computational design of CXCR4-dimer conformations with distinct stabilities

Inactive and active state open-dimer and closed-dimer models of the WT receptor served as starting templates for all design calculations performed using the implicit lipid membrane model of RosettaMembrane[28,44,45]. Positions at the interface of the two protomers were systematically scanned in silico (~20 positions, $20^{20}$ possible

combinations) to search for mutations that would stabilize the open-dimer or closed-dimer conformation without modifying significantly the stability of each monomer. This strategy ensured that designed mutations would solely affect the structural and functional properties associated with receptor dimerization. Hence, mutations were selected according to the quantity $\Delta E_{interface}$ *using the following equation:*

$$\Delta E_{interface} = (E_{interface})_{design} - (E_{interface})_{WT} \qquad (1)$$

where

$$E_{interface} = E_{dimer} - 2*E_{monomer} \qquad (2)$$

providing $\Delta E_{monomer} = (E_{monomer})_{design} - (E_{monomer})_{WT} \qquad (3)$

remained minimal.

Any designed mutation that had minimal effects on $\Delta E_{interface}$ (<1.0 REU) and/or significantly affected $\Delta E_{monomer}$ (>1.0 REU) was systematically discarded. After each step of sequence selection, the structure of the designed binding interface was refined and optimized using a Monte Carlo Minimization protocol sampling all conformational degrees of freedom.

The distribution between dimer conformations for the final selected designs and the associated functional effects on the binding to G protein versus β-arrestin were obtained by performing a final round of docking simulations where designed monomers were assembled into GPCR dimers and into complex with G proteins or β-arrestin as described above for the WT receptor.

### Modeling μOR active state dimer structures

Starting from the active state monomeric structure of μOR bound to the G protein Gi (PDB 6DDF), homodimer models of the WT receptor were obtained using the symmetric docking mode of RosettaMembrane described above using the same parameters than for CXCR4. Representative lowest-energy homodimer μOR models were selected to assemble Gi and β-arrestin complexes as described above for CXCR4. The bound Gi structure resolved in the 6DDF structure was used for docking onto μOR dimer models. Final models were selected and analyzed using the same unbiased geometric and energetic criteria as for CXCR4. The effect of the W5.34 A mutation was obtained by calculating the quantity $\Delta E_{interface}$ after assembling the mutated monomers into GPCR dimers as described in the computational design section.

### Calculation of dimerization propensity

The docking simulations performed using the software Rosetta do not reliably calculate free energies of protein associations because they neglect conformational and configurational entropies for example and just provide the enthalpy of a static structure. Nevertheless, differences in dimerization propensities between receptor variants can be estimated from the dimer binding energy calculated for the selected open and closed-dimer conformation as follows. In absence of free energies for the monomer and dimer species, we define a reference state, that of the lowest-energy primary dimer conformation of the WT receptor, i.e. the open dimer: $(\Delta E_{interface,\,O})_{WT}$. We first calculate the difference in dimer binding energies for each variant and conformation from WT as follows:

$$(\Delta\Delta E_{interface,\,Y})_X = (\Delta E_{interface,\,Y})_X - (\Delta E_{interface,\,O})_{WT} \qquad (4)$$

where X represents WT or any designed receptor variant and Y = O or C and corresponds to the open and closed conformation, respectively.

The Boltzmann factors describing the probability of a variant X to occupy the dimer state in a specific conformation $((PD_Y)_X$, dimerization

propensity) relative to WT can be derived as follows:

$$(PD_Y)_X = \exp(-(0.5((\Delta\Delta E_{interface,\,Y})_X)/RT) \qquad (5)$$

where the 0.5 factor roughly converts Rosetta Energy Units to kcal/mol. RT is the thermal scaling factor and equal to 0.593 kcal.mol$^{-1}$.

The sum of the Boltzmann factors for the open and closed conformation are calculated in the inactive and active state (reported in Supplementary Table 2) and provides an indication whether a variant has a lower or higher propensity to occupy the dimer state than WT.

### Alpha-Fold predictions

The Alphafold2-multimer[38] algorithm implemented by ColabFold[39] was applied to generate Alphafold2 models. The ColabFold implementation enables to provide custom structural templates to the program. Steric clashes in AF2 models of CXCR4 bound to Gi or β-arrestin were calculated using the ChimeraX software[46].

### Reagents and plasmids

CXCL12 was purchased from Cedarlane. Forskolin, isobutylmethyl xanthine (IBMX), AVP, and met-enkephalin were purchased from Sigma. The following plasmids were already described: HA-CXCR4[47], β-arrestin-2-LucII[48], β-arrestin-2-YFP[49], Gα$_{i1}$–91RLucII[47], Gα$_s$–117RLucII[50], Gα$_{oA}$–91RLucII[51], GFP10-G$_{\gamma1}$[52], GFP10-G$_{\gamma2}$[53], GFP10-EPAC-RLucII[54] and rGFP-CAAX[33]. The cloning of CXCR4-RLuc and CXCR4-YFP in pcDNA3.1 was previously described[11]. In the present study, CXCR4-RLuc and CXCR4-YFP were amplified and modified by PCR at the N-terminal end to add a myc epitope (EQKLISEEDL) or a HA epitope (YPYDVPDYA), respectively. Myc-CXCR4-RLuc and HA-CXCR4-YFP segments were then subcloned into pIREShyg3 (BsrG1/AflII) and pIRESpuro3 (NheI/AflII) respectively. The human μOR and V2R were amplified with a SNAP tag at their N-terminal (NEB) and subcloned in the pcDNA4/TO plasmid (Invitrogen). All the mutants were obtained by site-directed mutagenesis using the extension of overlapping gene segments by PCR technique and validated by sequencing.

### Cell culture and Transfections

Human Embryonic Kidney 293 T cells (HEK293T cells) were cultured using Dulbecco's Modified Eagle Medium (DMEM with L-glutamine from Wisent) supplemented with 10% vol/vol Fetal Bovine Serum (Wisent). The day before transfection, 600,000 cells were seeded in 6-well plates. Transient transfections were performed using Polyethylenimine 25 Kd linear (PEI, Polysciences) as transfection agent, with a 3:1 PEI:DNA ratio.

U87.MG cells stably expressing HA-CXCR4 and HA-CXCR4-W195$^{5.34}$L mutant (U87.CXCR4 and U87.CXCR4-W195$^{5.34}$L, respectively) were established by transfection of pIRES-HA-CXCR4 and pIRES-HA-CXCR4-W195$^{5.34}$L and subsequent cell sorting for equivalent surface expression levels using Alexa Fluor 488-labeled anti-HA antibody 1:1000 (Biolegend, clone 16B12,). U87 cells were grown in Dulbecco's modified Eagle medium (Themo Fischer Scientific) supplemented with 15% vol/vol fetal bovine serum and penicillin/streptomycin (100 Units/ml and 100 μg/ml) (Themo Fischer Scientific). U87.CXCR4 and U87.CXCR4-W195$^{5.34}$L cell lines were maintained under puromycin (0.5 μg/ml) selective pressure.

Cells were regularly tested for mycoplasma contamination (PCR Mycoplasma Detection kit, abm). If contamination was detected, cells were discarded and replaced from a frozen mycoplasma-free cell stock of lower passage.

### BRET measurements

Two different BRET configurations were used in this study: BRET$_{480-YFP}$ and BRET$_{400-GFP10}$. BRET$_{480-YFP}$ uses RLuc as energy donor and YFP as the acceptor (excitation peak at 488 nm) and coelenterazine-h (coel-h, Nanolight Technology) was used as the substrate (emission peak at

480 nm). BRET$_{400\text{-}GFP10}$ uses RLucII as energy donor and GFP10 as the acceptor (excitation peak at 400 nm) and coelenterazine-400a (coel-400a, Nanolight Technology) was used as the substrate (emission peak at 400 nm). Enhanced bystander BRET (ebBRET) uses RlucII as energy donor, rGFP as the acceptor and is detected using the BRET$_{480\text{-}YFP}$ configuration and Prolume Purple as the substrate (NanoLight Technology). BRET was measured with a Mithras LB940 multimode microplate reader (Berthold Technologies) equipped with a BRET$_{480\text{-}YFP}$ filters set (donor $480 \pm 20$ nm and acceptor $530 \pm 20$ nm filters) or a Tristar microplate reader equipped either with a BRET$_{480\text{-}YFP}$ filters set (donor $480 \pm 20$ nm and acceptor $530 \pm 20$ nm filters) or a BRET$_{400\text{-}GFP10}$ filters set (donor $400 \pm 70$ nm and acceptor $515 \pm 20$ nm filters). All the BRET experiments were performed at room temperature.

**CXCR4 dimerization.** Cells were transfected with HA-CXCR4-YFP and myc-CXCR4-RLuc, WT or mutant, and seeded in 96-well plates (Culturplate, Perkinelmer) coated with poly-L-ornithine (Sigma Aldrich) 24 h after transfection. The following day, cells were washed with Hank's Balanced Salt Solution (HBSS, Invitrogen) and incubated in HBSS supplemented with 0.1% BSA. Cells were treated with CXCL12 at the indicated times and concentrations. Coel-h (2.5 μM) was added 10 min before reading.

**G protein activation.** Cells were transfected with the receptor (CXCR4, V2R or μOR) and a three-component BRET-based biosensor: Gαi1-RlucII (CXCR4), Gαs117RlucII (V2R) or Gαo-RlucII (μOR) and Gβ1, and Gγ1-GFP10 (V2R and μOR) or Gγ2-GFP10 (CXCR4). BRET was then monitored as described above using coel-400a as a substrate. The dissociation of the Gα and Gβ/Gγ subunits after activation leads to a decrease in the BRET ratio.

**β-arrestin engagement (direct interaction).** Cells were transfected with CXCR4-Rluc and β-arrestin-2-YFP. BRET was monitored as described above using Coel-h as a substrate.

**β-arrestin engagement (ebBRET).** Cells were transfected with the receptor (HA-CXCR4, SNAP-V2R or SNAP- μOR), β-arrestin-2-RLucII, and CAAX-rGFP. BRET was monitored as described above using Prolume Purple (1.3 μM) as a substrate.

**cAMP accumulation.** Cells were transfected with HA-CXCR4 and the BRET-based biosensor GFP10-EPAC1-RlucII. BRET was then monitored as described above with the cells first washed with HBSS and then incubated in HBSS + 0.1% BSA containing 500 μM isobutylmethyl xanthine (IBMX), without or with 10 μM forskolin for 15 min, followed by agonist stimulation.

**CXCR4 phosphorylation**

HEK293 cells were seeded in six-well plates and transfected with either HA-CXCR4-WT or HA-CXCR4-W195L. 48 h later, cells were washed with phosphate-buffered saline (PBS) and serum starved in HBSS for 2 h. Cells were then stimulated with 200 nM CXCL12 or vehicle for 30 min before washing with ice-cold PBS and lysed with RIPA lysis buffer (50 mM Tris-HCl pH 7.6, 150 mM NaCl, 1% NP-40, 0.5% sodium deoxycholate, 0.1% SDS complemented with Halt™ Protease Inhibitor Cocktail (Thermo Fischer) and PhosSTOP phosphatase inhibitor (Roche)). Cell lysates were cleared by centrifugation at 14 000 rpm for 15 min at 4 °C and the supernatant was mixed with 2X SDS sample buffer. Samples were then analyzed by electrophoresis on a 10% SDS-polyacrylamide gel, transferred to a PVDF membrane, and immunoblotted with the following antibodies: pS324/pS325-CXCR4 phospho-CXC Chemokine Receptor 4 rabbit 1:1000 (7TM antibodies), anti-HA 3F10 rat monoclonal 1:1000 (Roche) and anti-GAPDH rabbit monoclonal 1:5000 (Cell Signaling) antibodies. Membranes were then washed and incubated with HRP-coupled secondary donkey anti-

rabbit IgG (GE healthcare) or goat anti-rat IgG (Sigma) antibodies, and the images were acquired and analyzed using a Chemidoc Imaging System (Bio-Rad). The unprocessed scans of the Western blots are presented in Supplementary Fig. 16.

**ERK phosphorylation assay monitored by HTRF**

U87, U87.CXCR4 and U87.CXCR4-W195$^{5.34}$L cells were seeded in 96-well plates ($1 \times 10^4$ cell/well). 72 h later, culture medium was replaced with FBS-free, phenol-red free DMEM. After 4-hour starvation, CXCL12 was added to cells at a final concentration of 10 nM and incubated for the indicated times. ERK phosphorylation was evaluated using a Homogenous Time-Resolved FRET (HTRF)-based Phospho-ERK (Thr202/Tyr204) cellular kit (Cisbio, 64AERPET). Cells were lysed for 30 min with the lysis buffer provided and incubated for 2 h with pERK1/2-specific antibodies conjugated with Eu$^{3+}$-cryptate donor and d2 acceptor at recommended dilutions. HTRF was measured with Tecan GENios Pro plate reader equipped with $612 \pm 10$ (donor) and $670 \pm 25$ (acceptor) filters. HTRF ratio was calculated as follows:

$$\text{Ratio} = \frac{A_{670}}{D_{612}} \times 10000 \qquad (6)$$

Where $A_{670}$ = emission at 670 nm (RFU) and $D_{612}$ = emission at 612 nm (RFU).

**Elisa**

To control for the cell surface expression of HA-CXCR4, HA-CXCR4-YFP, and myc-CXCR4-Rluc, and their respective mutant receptors, ELISA were performed in parallel of BRET experiments, using an antibody directed at the extracellular epitope (HA or Myc). 24 h after transfection, cells were seeded in 24-well plates coated with poly-L-ornithine. The day of the experiment, media was removed and a solution of PBS with 3.7% paraformaldehyde was added for 5 min. Cells were then washed 3 times with Phosphate-Buffered Saline (PBS). Blocking solution (PBS + 1% BSA) was added for 45 min then replaced by PBS + 1% BSA containing HA mouse monoclonal 12CA5 antibody 1:1000 (Santacruz) or Myc-tag rabbit 71D10 mAb 1:1000 (Cell Signaling) for 45 min. After antibody addition, cells were washed three times with PBS and incubated 45 min with PBS + 1% BSA containing an HRP-tagged sheep anti-mouse or donkey anti-rabbit IgG antibodies 1:2000 (GE healthcare). After labeling, cells were washed three times with PBS and incubated with SigmaFastOPD (SigmaAldrich) at room temperature. Reaction was stopped using 3 N HCl, supernatant transferred in a 96-well plate, and reading was performed using a Spectramax multimode microplate reader (Molecular Devices) at 492 nm.

**Flow cytometry**

Endogenous CXCR4 expression on the surface of HEK and U87 cells was monitored by flow cytometry using CXCR4-specific phycoerythrin-conjugated mAb 12G5 1:20 or the corresponding isotype control (R&D Systems) in a BD FACS LSR Fortessa cytometer (BD Biosciences). U87 was chosen as the cellular background for the absence of endogenous CXCR4 and ACKR3, as previously demonstrated[55,56]. U87 cells stably expressing the HA-tagged CXCR4 or variants thereof were obtained following puromycin selection and subsequent single-cell sorting using BD FACSAria II cell sorter (BD Biosciences). The equivalent surface expression level was verified using an Alexa Fluor 488-conjugated anti-HA-tag mAb 1:1000 (Biolegend, clone 16B12). Flow cytometry data were analyzed using FlowJo V10 software.

**Data and statistical analysis**

All data were analyzed using GraphPad Prism (GraphPad Software, Inc). Statistical significance between the groups was assessed with unpaired *t* test, a one-way ANOVA followed by Tukey's post hoc test, or a two-way ANOVA followed by Šídák's multiple comparisons test.

## Reporting summary

Further information on research design is available in the Nature Research Reporting Summary linked to this article.

## Data availability

The data supporting the findings in this study are present within the article and its Supplementary Information files, and are available from the corresponding authors upon request. All the biosensors can be obtained and used without limitations for academic non-commercial studies through regular Material Transfer Agreements and can be requested by email from Michel Bouvier. The following PDB entries were used for modeling: 3ODU, 4XT1, 6DDF, 7DW9, 4ZWJ, 3SN6, and 1GIA. Source data are provided with this paper.

## Code availability

Examples with commands/inputs/outputs/code for running the symmetry docking, clustering analysis, Gi & β-arresting docking, and Alphafold are provided in the github repository: https://github.com/barth-lab/QUESTS.

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

## Acknowledgements

This work was supported by a Swiss National Science Foundation grant (31003A_182263 and 310030_208179), a Novartis Foundation for medical-biological Research grant 21C195, a Swiss Cancer Research grant (KFS-4687-02-2019), a National Institute of Health grant (1R01GM097207), funds from EPFL, and the Ludwig Institute for Cancer Research to P.B., and a grant from the Canadian Institute for health Research (CIHR) (Foundation grant #148431) to M.B. M.J.S. and A.C. were supported by the Luxembourg National Research Fund (Pathfinder "Interceptor" 19/14260467, INTER/FWO "Nanokine" grant 15/10358798, INTER/FNRS grants 20/15084569, and PoC "Megakine" 19/14209621) and F.R.S.-FNRS-Télévie (grants 7.4593.19, 7.4529.19 and 7.8504.20). J.S.P. had studentships from the 'Groupe de Recherche Universitaire sur le Médicament' and 'la Faculté des Études Supérieures et postdoctorales de l'Université de Montréal'. R.E.J. was supported by a Marie Curie Postdoctoral Fellowship and received funding for this project from the European Union's Horizon 2020 research and innovation programme under the Marie Skłodowska-Curie grant agreement No 588412. B.S. holds a studentship from the 'Fond de Recherche du Québec–Santé' (FRQ-S). B.M. had a fellowship from the 'Fondation pour la Recherche Médicale (France). M.B. Holds a Canada Research Chair in Signal Transduction and Molecular Pharmacology. The authors thank Dr. Monique Lagacé for her critical reading of the manuscript.

## Author contributions

P.B., A.C., and M.B. designed the study; X.F., R.E.J., and P.B. performed the modeling and design calculations; J.S.P., B.M., B.S., M.S., M.H., N.D.B., and F.M.H. performed the experiments under the supervision of M.B, A.C., and M.J.S.; all authors analyzed the data; P.B., J.S.P., X.F., B.S., and M.B. wrote the manuscript. B.M., R.E.J., and B.S. made equal contributions.

## Competing interests

M.B. is the president of the scientific advisory Board of Domain Therapeutics, which licensed some of the BRET-based biosensors used in the present study for their commercial use. P.B. holds patents and provisional patent applications in the field of engineered T-cell therapies and protein design. The other authors declare no competing interests.
