## [Peer Review File · Nature Communications]

Computationally designed GPCR quaternary structures bias signaling pathway activationEditorial Note: This manuscript has been previously reviewed at another journal that is not operating a transparent peer review scheme. This document only contains reviewer comments and rebuttal letters for versions considered at *Nature Communications*.

REVIEWER COMMENTS

Reviewer #1 (Remarks to the Author):

The work presented by Barth and coworkers aims to use a combined approach that initially employs computational prediction of dimer interfaces of GPCRs to identify point mutations that can stabilize dimer interfaces in an open or closed state, followed by validation (mainly) via BRET assays. The problem that Barth and co. are tackling (GPCR dimerization and the effect on signaling through G proteins and beta-arrestin (BAR)) is a critical and significantly difficult one.

The most significant result is that they have identified a conserved motif among a subset of GPCRs that are known to dimerize along the TM4/TM5 interface that acts as a selectivity filter for signaling through BAR. This result was demonstrated for three GPCRs: CXCR4 chemokine receptor, mu opioid receptor (MOR), and the vasopressin receptor (V2R). In addition, Barth and coworkers extended their iPHOLD approach to include prediction of tertiary complexes (GPCR dimer + G protein or BAR), a critical development to the success of this work and one that could potentially reap long-term dividends for the GPCR structural biology and medicinal chemistry community.

While I agree with the overall conclusions of the study, I have several concerns about the experimental data that was presented. This may simply be attributable to the need for a more detailed explanation in the text, or it could be that the experimental design needs to be augmented.

1) bottom of p. 7, line 160: the dimerization propensities are calculated relative to the WT (Table S2). The propensity for L194R in the inactive state is > 7x larger than the WT, whereas the W195L is merely 1.76x larger than the WT. What is the significance of this large difference in propensity values? They were essentially presented as a general relationship.

2) line 175: " This increase in BRET can be interpreted as a change in conformation within dimers and a shift toward the closed dimer form, as suggested by our calculations, or as an increase in dimer population upon activation". This is intentionally ambiguous and comes across as an attempt to skirt around the issue that BRET lacks the resolution to unambiguously characterize this increase in signal. This issue needs to be addressed, as it starts to become a recurring theme throughout the rest of the study.

3) line 196: I don't disagree that the closed-dimer mutants show constitutively lower dimerization rates that increase upon agonist binding. However, the increase in dimerization of the W195L mutant is so drastically higher than the N192W mutant (Fig. 2a) -- even higher than the WT and all open dimer mutants -- that this needs to be addressed. Why is this response so much higher? What are the structural underpinnings of this difference, since the closed dimer point mutants are at the same interface (and 3 positions apart)?

4) line 199: "strong correlation" is a bit of an overstatement. The line fit (Fig. S7) was for 4 data points. Come up with a more robust approach to quantify this correlation or tone down the discussion.

5) line 240, discussion of ERK activation with respect to W195L closed dimer (Fig. 3C): please clarify this section. I either missed the connection or the connection wasn't explicitly made:

- W195L restores dimerization of CXCR4 upon agonist binding (Fig. 2a)
- W195L binds Gi as effectively as WT (Fig. 2b)
- W195L signaling (via cAMP decrease) is effectively the same as WT (Fig. 2c)
- W195L binding to BAR is almost non-existent (Fig. 3a)
- ERK signaling is dependent on BAR/Gi binding. Since BAR binding is already decreased with W195L, ERK activation should be decreased, which is what is observed (Fig. 3c). The discussion around this data is a circular argument -- without increased levels of BAR binding (which was already known), ERK activation cannot be high.
- This is fulcrum of the paper, and thus needs more convincing data/explanation of the data. W195L is clearly the most interesting mutation that was identified, as it exhibits non-conforming behavior compared to the WT (BAR binding) and to other closed mutants (dimerization upon activation). Would love to see some DEER measurements of the WT and W195L mutants here, but realize that this is most likely outside of the expertise of the group. A higher-resolution spectroscopic technique is really needed to identify how W195L is driving structural differences in the CXCR4 dimer.

6) line 283: appears there is an unnecessary sentence fragment here

7) discussion starting with line 296: the authors computational results for MOR (not detectable in complex with Go or BAR, Table S3) are not consistent with the experimental results (Fig. 5b). There is clearly a detectable fraction of MOR dimers of the W230A mutant that still bind Go (> 50%). The computational models should definitely be able to construct this. "ND" on MOR raises concerns that the computational approach is not as robust as the authors claim in light of the experimental data.

8) Fig. 5: suggest making the line styles consistent for both part A and B. The G protein and BAR lines are opposite for each segment (dashed vs. solid).

Reviewer #2 (Remarks to the Author):

The authors report a computational approach which identifies the mutants which stabilize different oligomeric conformations of GPCR proteins, and experimentally show that their designs have different functionally on the example of CXCR4 with respect to recruiting of beta arrestin in one dimer conformation.

Second they identify that the residue , where mutation strengthening the "new" dimer occurs, is highly evolutionary conserved in many other GPCRs and hence the authors claim that they uncovered a mechanism of conformational switch which modulates function of multiple GPCRS.

Overall this is an interesting paper with reasonable methods and experimental validation. It would be good if the authors would include little bit more data and information to support the actual 3D structures they predict, since the authors jump directly to functional validation.

Specifically the authors provided the dimer structures in the supplement, but not the complexes with beta arrestin which (they show it only on the figure). Also would be good to predict the models of the dimers and the complex with AlphaFold 2 multimer (AF). If AF2 doesn't produce meaningful models - the authors can run AF 2 with their models as the starting template and see wether AF considers those models reliable. Same can be done with their CXCR4 beta arrestin model. It might not work, but if it does this would further support the paper. Also the authors can attempt to model the oligomers of the other GPCRs they discuss in the paper with AF to further support the story.

REVIEWER COMMENTS

Reviewer #1 (Remarks to the Author):

The work presented by Barth and coworkers aims to use a combined approach that initially employs computational prediction of dimer interfaces of GPCRs to identify point mutations that can stabilize dimer interfaces in an open or closed state, followed by validation (mainly) via BRET assays. The problem that Barth and co. are tackling (GPCR dimerization and the effect on signaling through G proteins and beta-arrestin (BAR)) is a critical and significantly difficult one.

The most significant result is that they have identified a conserved motif among a subset of GPCRs that are known to dimerize along the TM4/TM5 interface that acts as a selectivity filter for signaling through BAR. This result was demonstrated for three GPCRs: CXCR4 chemokine receptor, mu opioid receptor (MOR), and the vasopressin receptor (V2R). In addition, Barth and coworkers extended their iPHOLD approach to include prediction of tertiary complexes (GPCR dimer + G protein or BAR), a critical development to the success of this work and one that could potentially reap long-term dividends for the GPCR structural biology and medicinal chemistry community.

While I agree with the overall conclusions of the study, I have several concerns about the experimental data that was presented. This may simply be attributable to the need for a more detailed explanation in the text, or it could be that the experimental design needs to be augmented.

We thank the reviewer for the overall positive assessment of our work.

1) bottom of p. 7, line 160: the dimerization propensities are calculated relative to the WT (Table S2). The propensity for L194R in the inactive state is > 7x larger than the WT, whereas the W195L is merely 1.76x larger than the WT. What is the significance of this large difference in propensity values? They were essentially presented as a general relationship.

The dimerization propensities are directly derived from the dimer interface energies calculated by Rosetta and reported in **Supplementary Table 1**. As shown in **Fig. 2a**, the designed Arg at position 194 creates two strong hydrogen bonds with the other protomer in the open inactive state conformation. These interactions stabilize the open and closed dimer forms in the inactive state by 4.85 Rosetta Energy Units as compared to the WT receptor. When translated into a Boltzmann weight as described in the methods, this leads to a large increase in dimerization propensity. On the other hand, the L195 mutation creates novel VDW interactions stabilizing the closed active state conformation (**Fig. 2b**). However, because the mutation also significantly destabilizes the open dimer form, we predict a net stabilizing effect on the active state dimer of only 0.92 Rosetta Energy Units (hence the lower increase in propensity). While the differences in propensity can be directly linked to distinct structural interactions in our designs, it should be noted that our calculations do not provide true free energy estimates of dimerization. Since the propensities are calculated from differences in dimer structure stability between variants, we do not expect these numbers to quantitatively predict true dimerization equilibria properties. Nevertheless, except for L194R, we observe a qualitative trend between the propensities and experimental measurements which suggest that designed structural interactions may impact the dimerization propensity as intended by the design calculations. We clarified this point in the manuscript page 7 (where we discuss calculated dimerization propensities) and in page 9 (where we compare these predictions to the BRET data).

2) line 175: " This increase in BRET can be interpreted as a change in conformation within dimers and a shift toward the closed dimer form, as suggested by our calculations, or as an increase in dimer population upon activation". This is intentionally ambiguous and comes across as an attempt to skirt around the issue that BRET lacks the resolution to unambiguously characterize this increase in signal. This issue needs to be addressed, as it starts to become a recurring theme throughout the rest of the study.

The reviewer is right that BRET cannot distinguish between a change in conformation resulting in a closed dimer or an increase in the dimer/monomer ratio. Yet, the BRET changes observed are entirely consistent with the predicted outcomes of the computational predictions since receptor activation is predicted to favor both an increase in dimerization and the closed dimer conformation (see for example WT receptor in **Supplementary Table 1**). To experimentally ascertain that the change in dimer conformation occurs and contribute to the BRET change, the 3D structures resolved by X-ray or Cryo-EM of the WT and mutant forms of the receptor in the presence and absence of ligands and in complex with β arrestin and G proteins would be required. Even such structural information that goes beyond the scope of the present manuscript, would not allow to directly address the dynamic equilibrium between monomer and dimer forms.

To make this point more explicitly, we modified the sentence as follows: 'This increase in BRET can be interpreted as a change in conformation within dimers and a shift toward the closed dimer form or as an increase in dimer population upon activation that were both suggested by our calculations (**Supplementary Table 1**). Although both phenomenon most likely contribute to the increase, their relative contribution cannot be determined from the BRET data or, to our knowledge, any other experimental approach (Page 9, 1st paragraph).

3) line 196: I don't disagree that the closed-dimer mutants show constitutively lower dimerization rates that increase upon agonist binding. However, the increase in dimerization of the W195L mutant is so drastically higher than the N192W mutant (Fig. 2a) -- even higher than the WT and all open dimer mutants -- that this needs to be addressed. Why is this response so much higher? What are the structural underpinnings of this difference, since the closed dimer point mutants are at the same interface (and 3 positions apart)?

Despite being only 3 residues apart, N192 and W195 do not point to the same direction and are not part of the same interface because W195 is on TM helix 5 while N192 belongs to the extracellular loop 2. In addition to the local effects of the mutation on the inter-protomer interactions described in **Figure 2a and b**, W195L enables better overall packing of the extracellular side of the dimer interface. The smaller size of the side chain at position 195 enables closer contacts between the 2 protomers specifically in the active state which result in the largest increase in buried surface area upon dimerization ($\Delta\text{SASA} = 3216 \text{ \AA}^2$) among all variants. In comparison, the ΔSASA calculated for the N192W in the active state (3063 \AA^2) is only slightly larger than WT (3026 \AA^2). We created a new figure panel (**Fig. 2e**) reporting this structural analysis which is described in page 7.

4) line 199: "strong correlation" is a bit of an overstatement. The line fit (Fig. S7) was for 4 data points. Come up with a more robust approach to quantify this correlation or tone down the discussion.

We agree with the reviewer that the semantic was not appropriate. We updated the manuscript with the following sentence: "... consistent trend between predicted closed dimer stabilization and increase in Δ BRET upon activation " (page 10, 2nd paragraph).

5) line 240, discussion of ERK activation with respect to W195L closed dimer (Fig. 3C): please clarify this section. I either missed the connection or the connection wasn't explicitly made:

- W195L restores dimerization of CXCR4 upon agonist binding (Fig. 2a)
- W195L binds Gi as effectively as WT (Fig. 2b)
- W195L signaling (via cAMP decrease) is effectively the same as WT (Fig. 2c)
- W195L binding to BAR is almost non-existent (Fig. 3a)
- ERK signaling is dependent on BAR/Gi binding. Since BAR binding is already decreased with W195L, ERK activation should be decreased, which is what is observed (Fig. 3c). The discussion around this data is a circular argument -- without increased levels of BAR binding (which was already known), ERK activation cannot be high.

The points listed by the reviewer are exact. To clarify the point that we wanted to address by measuring the ERK activation, we changed the last concluding sentence of the paragraph. These experiments were done to assess whether the impact of the mutation on the β arrestin recruitment were resulting into the expected functional outcome at the signaling level. This is indeed the case as is now explicitly stated in the added sentence that reads as follows: 'These data show that the designed change in dimerization resulted into a functional outcome at the signaling level reflected by a blunted ERK response, a result that is consistent with the reduced β arrestin recruitment observed for this mutant.' (Page 12, 1st paragraph).

6) line 283: appears there is an unnecessary sentence fragment here

We removed the sentence fragment.

7) discussion starting with line 296: the authors computational results for MOR (not detectable in complex with Go or BAR, Table S3) are not consistent with the experimental results (Fig. 5b). There is clearly a detectable fraction of MOR dimers of the W230A mutant that still bind Go (> 50%). The computational models should definitely be able to construct this. "ND" on MOR raises concerns that the computational approach is not as robust as the authors claim in light of the experimental data.

We apologize for the confusion as N.D. meant not determined. As we focused our analysis on the shift in the population of open and wide-open conformations, we did not initially report the binding energies of the mOR variants to G proteins and arrestin in **Supplementary Table 3**. We performed additional docking calculations of Go and arrestin to the different active state conformations of WT and W230A mOR and updated **Supplementary Table 3** accordingly. In summary, we observe similar binding energies of the WT and mutant wide-open conformations to Go or arrestin. By contrast, the binding of the open conformation to Go and especially to arrestin is affected by the W230A mutation, resulting in decreased binding energies as compared to WT (i.e. by 0.37 and 0.87 REU for Go and arrestin, respectively).

Our predictions are consistent with the experimental BRET measurements of arrestin recruitment: first, our calculations suggest that the mutation induced a population shift of the receptor dimers toward the wide-open conformation (i.e. by 1.8 REU in dimerization energy) which prevents optimal binding to the finger loop of arrestin and consequently displays a substantially weaker interaction to arrestin than the open conformation (see new **supplementary Figures 11, 12**). Second, our calculations suggest that the mutation decreases also the binding of arrestin to the open conformation. Overall, the combination of these 2 mutational effects is predictive of an overall decrease in arrestin binding which is consistent with the experiments. Concerning the milder effects

of W230A on the recruitment of Go, our calculations are also consistent with the experimental observations and suggest smaller mutational effects since the binding propensity to Go of the open and wide-open conformations are closer to each other than for arrestin. We refer to the new mOR data in the main text pages 13-14.

8) Fig. 5: suggest making the line styles consistent for both part A and B. The G protein and BAR lines are opposite for each segment (dashed vs. solid).

The figure has been changed according to the suggestion.

Reviewer #2 (Remarks to the Author):

The authors report a computational approach which identifies the mutants which stabilize different oligomeric conformations of GPCR proteins, and experimentally show that their designs have different functionality on the example of CXCR4 with respect to recruiting of beta arrestin in one dimer conformation.

Second, they identify that the residue, where mutation strengthening the “new” dimer occurs, is highly evolutionary conserved in many other GPCRs and hence the authors claim that they uncovered a mechanism of conformational switch which modulates function of multiple GPCRS.

Overall this is an interesting paper with reasonable methods and experimental validation.

We thank the reviewer for the overall positive assessment of our study.

It would be good if the authors would include little bit more data and information to support the actual 3D structures they predict, since the authors jump directly to functional validation. Specifically, the authors provided the dimer structures in the supplement, but not the complexes with beta arrestin which (they show it only on the figure).

We now provide all arrestin bound models in the **Supplementary figures 3,4,11,12**.

Also, would be good to predict the models of the dimers and the complex with AlphaFold 2 multimer (AF). If AF2 doesn't produce meaningful models - the authors can run AF 2 with their models as the starting template and see whether AF considers those models reliable. Same can be done with their CXCR4 beta arrestin model. It might not work, but if it does this would further support the paper. Also, the authors can attempt to model the oligomers of the other GPCRs they discuss in the paper with AF to further support the story.

We thank the reviewer for suggesting this comparison with AF2. We did run AF2 multimer on our complexes. While AF2 was not able to predict the CXCR4 dimer conformation on its own, it did recapitulate our predicted conformation when given our model as template (**Supplementary Fig. 13**). AF2 predictions of mOR dimers converged also to our modeled conformations when provided a template (**Supplementary Fig. 13**). We also predicted a V2R dimer conformation using AF2 which involves high density of contacts at the intracellular side of the receptor (**Supplementary Fig. 13**). Overall, our results suggest that AF2 might not yet provide very reliable and accurate predictions of class A GPCR dimer structures on its own, but, given the template, can recapitulate our predicted mode of self-association.

We obtained much better agreement with G-protein and beta-arrestin bound complex structures. Even without template, AF2 recapitulated our Beta-arrestin-2 bound CXCR4 models. All AF2 models predicted the orientation of the Beta-arrestin-2 and the main interactions between the arrestin finger loop and the receptor observed in our model (**Supplementary Fig. 14**). Importantly, AF2 predicted the same steric hindrance between arrestin and the receptor in the “closed-state” dimer that we identified using our approach (**Supplementary Fig. 15**). Lastly, while the AF2 predicted Gi-orientation in the Gi-bound CXCR4 complex was slightly different than in our models, the bound conformation of the G-protein and interactions with the receptor were not sensitive to the dimer conformation (**Supplementary Fig. 15**), consistent with our modeling calculations.

Overall, the AF2 models support the quaternary structure-based mechanism of biased signaling that we uncovered and report in this study. We updated the main text with these additional data page 15.

REVIEWERS' COMMENTS

Reviewer #1 (Remarks to the Author):

I appreciate the time and care taken to address my concerns with the initial submission. The revised manuscript has addressed these concerns, and I have no further criticisms.

Reviewer #2 (Remarks to the Author):

In the revised manuscript the authors improved and strengthened the structural aspect of the story by introducing AF 2 multimer structures, as was suggested. I feel it is appropriate for publication.